# A Novel Ultrasonic TOF Ranging System Using AlN Based PMUTs

**DOI:** 10.3390/mi12030284

**Published:** 2021-03-08

**Authors:** Yihsiang Chiu, Chen Wang, Dan Gong, Nan Li, Shenglin Ma, Yufeng Jin

**Affiliations:** 1School of Electronic and Computer Engineering Bldg. A, Peking University Shenzhen Graduate School, University Town, Xili, Nanshan, Shenzhen 518055, China; 1601111224@pku.edu.cn (Y.C.); wangchen2019@pku.edu.cn (C.W.); 1701213526@sz.pku.edu.cn (N.L.); 2Department of Mechanical & Electrical Engineering, School of Aeronautics and Astronautics, Xiamen University, Xiamen 361005, China; 19920171150940@stu.xmu.edu.cn

**Keywords:** aluminum nitride, piezoelectric micromachined ultrasonic transducers, ranging, time of flight (TOF), time to digital converter circuit (TDC)

## Abstract

This paper presents a high-accuracy complementary metal oxide semiconductor (CMOS) driven ultrasonic ranging system based on air coupled aluminum nitride (AlN) based piezoelectric micromachined ultrasonic transducers (PMUTs) using time of flight (TOF). The mode shape and the time-frequency characteristics of PMUTs are simulated and analyzed. Two pieces of PMUTs with a frequency of 97 kHz and 96 kHz are applied. One is used to transmit and the other is used to receive ultrasonic waves. The Time to Digital Converter circuit (TDC), correlating the clock frequency with sound velocity, is utilized for range finding via TOF calculated from the system clock cycle. An application specific integrated circuit (ASIC) chip is designed and fabricated on a 0.18 μm CMOS process to acquire data from the PMUT. Compared to state of the art, the developed ranging system features a wide range and high accuracy, which allows to measure the range of 50 cm with an average error of 0.63 mm. AlN based PMUT is a promising candidate for an integrated portable ranging system.

## 1. Introduction

Various piezoelectric materials, such as polyvinylidene fluoride (PVDF) [1], piezoelectric lead zirconate titanate (PZT) [2], zinc oxide (ZnO) [3] and aluminum nitride (AlN) [4], have been widely applied in piezoelectric micromachined ultrasonic transducers (PMUTs) and bulk ultrasonic transducers. Although PVDF is flexible with good processibility and fast dynamic response, mechanical deformation is required to covert the non-polarized phase to polarized phase. Therefore, it is hard to be integrated in the conventional micro processing. The disadvantages of PZT lie in its high deposition temperature (e.g., 600 °C) and high voltage required to be polarized. Although processing temperature is low, ZnO is poorly compatible with CMOS processing due to its proneness to chemical reaction with acid and base and the pollution problems during CMOS processing owing to fast diffusibility of zinc ions. In comparison, AlN can be deposited at low temperatures (<400 °C) and is compatible with complementary metal oxide semiconductor (CMOS) process [5,6]. Although AlN has a lower piezoelectric constant than that of PZT, AlN based PMUT is considered to be able to achieve comparable performance because of its lower dielectric constant than that of PZT. In the last decade, AlN PMUT has aroused an increasing global research interests and breakthrough has been achieved in its commercialization for ranging sensing. For instance, application-oriented R&D interest is increasing in recent years. In 2009, Shelton et al. [7], presented an AlN based PMUT with a film radius of 175, 200 and 225 μm operating in the 200 kHz range for short-range air coupling ultrasonic applications. When 1V input voltage was used to excite at a resonant frequency of 220 kHz, a film deflection of 210 nm was achieved. In 2014, Lu et al. [8], presented a high-frequency (10–55 MHz), fine-pitch (45–70 μm) PMUT array based on a customized silicon-on-insulator (SOI) wafer with buried cavities for medical imaging., The fabricated 9 × 9 array samples made of 40 μm diameter PZT PMUTs had a 3.4 MHz bandwidth at the center frequency of 10.4 MHz and measured pressure sensitivity of 2 kPa/V at a distance of 1.25 mm. In 2015, Horsley et al. [9], proposed an AlN PMUT for human-machine interfaces. In 2017, Goh et al. [10], proposed an AlN PMUT based ultrasonic measurement method for stainless steel tubes.

With the continuous development and wide application of computer, automation and industrial robots, the ranging technology, such as radar ranging [11], infrared ranging [12], laser ranging [13] and ultrasonic ranging [14], is becoming critical to ensuring accurate positioning. Compared with other ranging methods, ultrasonic ranging is not affected by the transparency and color of the target object, is insensitive to environmental noise, and can be used in direct sunlight. Besides, because of the relatively low speed of sound, ultrasonic transducer based ranging technology mitigates the high-speed electronics requirements confronted by RF (radio frequency) and optical ranging. Hence, ultrasonic ranging has the characteristics of low power consumption, which makes it an attractive alternative at short-range ranging (<10 m).

The traditional ultrasonic rangefinders use bulk piezoelectric ceramic with high output power. However, the mismatch of the acoustic impedance with air results in poor conversion efficiency between the electric field and the sound field [15]. Besides, their bulky size limits their use in portable devices. AlN has a quartzite crystal structure with a good piezoelectric effect along its c-axis orientation. It does not need to be polarized like lead PZT [4], and the AlN membrane has a lower deposition temperature than PZT (the former is less than 400 °C, and the latter is 600 °C), which results in its better compatibility with standard CMOS processes. Although the piezoelectric constant of AlN is not as high as PZT, its lower dielectric constant makes it possible to achieve performance comparable to PZT. Therefore, AlN based PMUT is a promising candidate for an integrated portable ranging system.

Regarding to ultrasonic ranging, either frequency modulated continuous wave mode (FMCW) or pulse echo mode (PE) is used. FMCW mode includes binary frequency shift keying (BFSK) [16], two frequencies continuous wave (TFCW) [17] and multi-frequencies continuous wave (MFCW) [18]. PE mode includes time of flight (TOF) [19]. Theoretically, frequency modulated continuous wave (FMCW) does not have dead zone while it is not the case for the PE mode [20]. However, measuring range of FMCW mode is short, and it is difficult to do Doppler coupling or isolate the transmitter and receiver. In comparison, the traditional flight (TOF) methods suffer from high levels of systematic errors due to degradation of the amplitude of received signal. To address this issue, the ultrasonic signal received by the PMUT is amplified by the high pass filter (HPF) and then being read by analog to digital converter (ADC) [14].

In this paper, two pieces of PMUTs are applied. One is used to transmit and the other is used to receive ultrasonic waves. The time to digital converter circuit (TDC), correlating the clock frequency with sound velocity, is utilized for range finding via TOF calculated from the system clock cycle. A prototype is fabricated by integrating two AlN PMUTs with an ASIC chip that is designed and fabricated on a 0.18 μm CMOS process. Its performance is studied experimentally. Compared to state of the art, the developed time to digital converter circuit (TDC) ranging system features a wide range and high accuracy.

## 2. Design of the Ranging System

The block diagram of the proposed ranging system is shown in Figure 1. The system consists of two PMUTs and a transceiver circuit. The ultrasonic transducer consists of a square AlN piezoelectric film sandwiched between two Mo electrodes suspended with buried vacuum cavity in the substrate. The voltage applied to the two electrodes forms an electric field in the thickness direction causing a transverse stress, which generates an acoustic wave from out-of-plane bending of the membrane. Similarly, the incident pressure wave causes the membrane to bend and generates a charge on the electrodes for receivers. When the PMUT is stimulated at a resonant frequency, the electromechanical conversion efficiency will be maximized.

The resonant frequency of the PMUT has a great influence on the ranging range and resolution. The resonance frequency of 100 kHz is chosen for the PMUT for modeling. If the PMUT is modeled as a rectangular membrane of uniform material with fully clamped boundaries (dimension: L_x_ and L_y_), the modal frequencies of the membrane are defined by Equation (1) [21]:(1)$$f_{m,n}=\frac{1}{2}\times \sqrt{\frac{T}{\sigma }}\times \sqrt{\frac{m^{2}}{L_{x}{}^{2}}+\frac{n^{2}}{L_{y}{}^{2}}}\;m,n=1,\;2,\;3,\ldots$$
where T is the surface tension, and σ is the area density. If $L_{x}=L_{y}=L$, the resonant frequency in the first mode (m = 1, n = 1) is as follows:(2)$$f_{0}=\frac{1}{2L}\sqrt{\frac{2T}{\sigma }}$$

It can be seen from Equation (2) that as the length of the side increases, the resonant frequency of PMUT decreases.

As shown in Figure 1a, the circuit used to transmit and receive signal from PMUTs fabricated by TSMC (Taiwan Semiconductor Manufacturing Company, Ltd., Taiwan, China) 0.18 µm CMOS process consists of four main portions: a charge pump circuit (CP), a digital unit (DU), a transmitter circuit (TX) and a receiver circuit (RX). As shown in Figure 1b, the CP will rise the voltage of 1.8 V to 32 V needed to drive the level shifter (LSF). The oscillator (OSC) will generate a system clock (system OSC) of 1.73 MHz and a drive clock (TX OSC) of 97 kHz. The voltage of TX OSC will rise to 32 V by LSF, which is then amplified by the TX driver to stimulate the PMUT. Next, the PMUT transforms the driving signal to acoustic wave and receives the reflected signal after the time of flight (TOF). The goal of the receiver circuit is to amplify and detect the received echo signal, and send it to digital part to derive the distance info from TOF. The digital unit will handle the control of the ranging process and support the setting adjustment and status reported through the I2C interface. The calibration functions are designed in this chip as well, and the results are stored in the OTP (one time programmable). The transceiver circuit only requires a single 1.8 V supply source and can be controlled via I2C interface. HV charge pump and clock generator are all integrated without adding external components. An interrupt output pin is arranged to minimize the polling bandwidth/burden of baseband chip. Once the proximity detection value is beyond the lower or upper thresholds, the interrupt is asserted to alert the host about the situation.

The distance (L) between the transducer and the object is figured out by TOF using ultrasound’s echo and receive roundtrip time Δt using Equation (3).
(3)$$L_{\min}=\frac{C_{\mathrm{us}}}{2f_{\mathrm{sys}}}$$
where $c_{\mathrm{us}}$ is the velocity of ultrasound in air and $f_{\mathrm{sys}}$ is the frequency of system. It is worth noting that sound velocity in air is not constant, which is affected by environment condition such as humidity and temperature. High humidity environment can quickly dampen the ultrasound, resulting in a very short transmission distance. The relationship between sound velocity and temperature can be described in Equation (4):(4)$$c_{\mathrm{us}}=\left(331.3+0.606\times T\right)\;m/s$$
where T is absolute temperature in unit of Celsius. The transmission speed of ultrasound in air at 25 °C is 346 m/s, which gives the ultrasound one–way travel time of 2.89 μs for 1 mm transmission distance. As shown in Figure 1a, the total travel distance of ultrasound from transmitter to receivers is twice that of the distance between the PMUT and the object (1 mm here). Hence, the total travel time of ultrasound is 5.89 μs, which corresponds to the system frequency of 1.73 MHz and minimum distance resolution of 0.1 mm using Equation (3). A 3D finite elements model of the PMUT was created using COMSOL Multiphysics software (Version 5.0, COMSOL Co. Ltd., Stockholm, Sweden) to study the mode shape and the time-frequency characteristics. The first vibration mode of the PMUT is shown in Figure 2a. PMUT operating in the first mode has maximum amplitude, and thus is more suitable for ranging applications. Figure 2b,c shows the simulation result of frequency characteristics of the PMUT. It can be seen that the resonant frequency is 100.17 kHz when the side length is 3500 μm, and the maximum displacement is about 75 μm at resonant frequency.

The propagation process of ultrasound is shown in Figure 2d. Starting from t=0 s, the film begins vibrating and radiating ultrasound outward with a fan angle of 180° (Figure 2(di–diii)). At t=120 μs, a part of the ultrasound bypasses the baffle and propagates downward, then reaches the receiver before the reflected waves (Figure 2(div,dv)). Finally, ultrasounds reach the reflector and reflects downward until they are detected by the receiver (Figure 2(dvi,dvii)).

Figure 2e,f show the simulated results of near-field and far-field sound pressure versus time. As seen from the Figure 2e, the envelopes of the ultrasonic waves at different points are very similar, their frequency is also the same, but the amplitude of the waves isometrically decrease, and the phase also moves equidistantly to the right. It is worth mentioning that the time difference of arrival (TDOA) of the three waves Δt is approximately $6\times 10^{-7}\;s$, and the distance difference Δs=v·Δt=0.2076 mm, where v=346 m/s is sound speed at room temperature, is exactly equal to the distance between two points. This shows that the ultrasonic wave is basically not affected by the external environment and is radiating out without distortion. For far-field sound pressure in Figure 2f, except for the 10 cm point, the envelopes of the other two waves have changed and the amplitudes will no longer attenuate after reaching the maximum, which may be caused by the superposition of the original wave and the reflected wave.

The received voltage of the receiver when the sensing distance is 7 cm and 8 cm is shown in Figure 2g,h, respectively. The ultrasounds shown in Figure 2(div,dv) that bypass the baffle and reach the receiver are the waves on the left of the red dashed line, and the right is the reflected waves for sensing. Therefore, the received voltage is the superposition of two waves, and the minimum amplitude between the two waves is the arrival time of the reflected waves. We can calculate the measured distance $s_{\left(g\right)}=v\cdot t_{\left(g\right)}/2=7.0065\;\mathrm{cm}$ and $s_{\left(h\right)}=v\cdot t_{\left(h\right)}/2=8.0099\;\mathrm{cm}$ from the arrival time. Since the results were very close to the actual distance, it shows that the method has extremely high accuracy.

## 3. System Implementation

### 3.1. Fabrication and Characterization of PMUTs

The steps to fabricate a PMUT device are shown in Figure 3a. The process starts with a bare silicon wafer with a thickness of 200 μm, followed by a layer of 1 μm SiO_2_ deposition using chemical vapor deposition (CVD). Afterwards, 0.1 μm-thickAlN seeds layer is deposited by atomic layer deposition (ALD) to ensure the formation of the (002) crystal orientation of AlN during the subsequent deposition Then, 0.1 μm-thick bottom Mo layer, 1 μm-thick AlN layer and 0.1 μm-thick Mo layer are deposited by physical vapor deposition (PVD) and patterned using SiO_2_ as hard masks, as shown in Figure 3(aIV–aVII). Subsequently, PE-SiO_2_ is deposited (Figure 3(aVIII)) and patterned (Figure 3(aIV)). Deposition of the top Al/Cu connection line and pads is shown in Figure 3(aX). Finally, a trench is etched from the back of the silicon wafer by reactive-ion etching (DRIE) to release the membrane structure, and a protective material is used to protect the front side membrane from damage as shown in Figure 3(aXI). In order to increase the sound pressure of the PMUT in the air, the bottom silicon is bonded to the glass using an anodic bonding process, and a vacuum cavity is formed by evacuation as shown in Figure 3(aXII). The top-view of PMUT is shown in Figure 3(bi) by optical microscopy (OM). The fabricated device is a 3500 μm-long square membrane with a cavity having dimensions of 2800 μm × 2800 μm × 167 μm. The top and bottom electrodes of the PMUT are connected to the signal and ground terminals via aluminum wires, respectively. Figure 3(bii) shows the secondary electron microscopy (SEM) imaging of the membrane with a close-up detail of the thickness of each layer. 

Two PMUTs with similar frequencies are selected as the transmitter and receiver for better ranging performance. As shown in Figure 3c,d, the impedance characteristics of two PMUTs are measured using an impedance analyzer (Agilent Technologies 4294A, Santa Clara, CA, USA). It can be seen that the resonant frequencies of the transmitter and receiver are 97 kHz and 96 kHz, which are very close to the simulation results. The electromechanical coupling coefficient $k_{\mathrm{eff}}^{2}$ can be derived by resonant frequency $f_{r}$ and anti-resonant frequency $f_{a}$ through Equation (5) [22]:(5)$$k_{\mathrm{eff}}^{2}=\frac{f_{a}{}^{2}-f_{r}{}^{2}}{f_{r}{}^{2}}$$

The calculated electromechanical coupling coefficients of the transmitter and receiver are 2.0% and 2.6%, respectively. Due to errors in the manufacturing process, it is difficult to ensure that the resonant frequencies and electromechanical coupling coefficients of the two PMUTs are identical.

### 3.2. System Clock Design and Implementation

The oscillator is used to create a main/system clock which is applied to ranging controller and TOF calculation. In order to easily represent or derive the distance result from TOF, the intuitive way is to correlate the clock frequency with sound velocity. The equation of sound speed is defined in Equation (4). At 25 °C, it can be derived that the TOF is 5.78 μs/mm. Adopting 1.73 MHz as the clock frequency of TOF calculator, we can get 0.1 mm resolution in theory, which is reasonable in the usage of ranging controller.

Since relaxation oscillator has the advantages of supply insensitive clocking, low power consuming and compatible with frequency trimming, it is adopted to generate the clock of the ranging system, as shown in Figure 4a. The oscillation frequency is derived using Equation (6):(6)$$f_{\mathrm{osc}}=\frac{I_{B}}{2C_{\mathrm{RAMP}}\times V_{\mathrm{REF}}}$$

In our design, $I_{B}=4\;\mu A$, $C_{\mathrm{RAMP}}\;\sim \;2\;\mathrm{pF}$, $V_{\mathrm{REF}}\;\sim \;0.5\;V$ were selected to obtain the desired clock frequency. Through simulation, the effects of different V_DD_ on the clock frequency variation were studied.

At a temperature of 25 °C, when the power supply V_DD_ rises from 1.62 V to 1.98 V, the clock frequency gradually decreases with a variation of 1.330% for post-sim, and the simulation results are shown in Figure 4b. The green dashed line indicates the pre-simulation results, while the red solid line indicates the post-simulation results. The former is higher than the latter because the post-sim considers the influence of parasitic capacitance.

Four chips were selected as samples for testing. The measurement results are summarized in Figure 4c. It can be seen from the experimental results that as V_DD_ increases, the clock frequency gradually decreases, and the tendency is the same as the simulation results. After frequency adjustment, the calibrated frequency can reach about 1.73 MHz, and the frequency error after calibration is well within 0.5% of the design target.

### 3.3. Charge Pump (CP)

The charge pump circuit is utilized to rise the voltage of 1.8 V to 32 V to drive the level shifter (LSF). It is a DC-DC converter that uses a capacitor as an energy storage element to generate an output voltage greater than the input voltage. When the drive voltage is 32 V, the PMUT can be fully activated. The output voltage of the CP is affected by the frequency and load. Through experimental measurement, the relationship between the output voltage and frequencies/load is determined, as shown in Figure 5a. The black line indicates the change of output voltage with frequency when there is no load, while the red line indicates the case when the load is 5.6 MΩ. It can be seen that the output voltage with load is lower than that without load, but the output voltage is positively related to frequency, from which can be estimated that the output voltage is about 29 V when the frequency is 97 kHz, as shown by the dotted line in the Figure 5a. Low output voltage is due to the parasitic cap, which will be improved in the future.

### 3.4. Transmitter (TX)

The two terminals of the circuit in this design were connected to the transmitter and receiver, respectively. As shown in Figure 5b, the TX circuit includes three blocks of TXOSC, Charge-Pump (CP), and HV Output Driver. Firstly, TXOSC with VREF circuit provides the internal voltage reference and generated clock supports charge-pump, CP5_CLK, CP32_CLK, and TXDATA of the HV output driver, respectively. It can provide the frequency range from 40 kHz to 30 MHz, and the frequency of TXOSC used in this work was 97 kHz.

Then, Charge-Pump block pumped high voltage from 1.8 V to 32 V with two blocks, one was 1.8 V to 5 V and another was 5 V to 32 V, for output driver to stimulate the transmitter. The operation frequency was 2 MHz for CP5V and 60 kHz for CP32V, respectively. Relative bypass capacitor for CP5V was an internal capacitor of 450 pF or co-package with external capacitor of 1 nF and CP32V was used with an external bypass capacitor of 1 nF or 10 nF.

Finally, HV Output Driver drives PMUT sensor with TXDATA with operation frequency and voltage magnitude. Figure 5b describes in detail the function block diagram of TX, and Figure 5c shows the signal simulation of the driver circuit.

### 3.5. Receiver (RX)

There are multiple receiver implementation solutions. In this design, pre-amplifier and echo comparators were adopted in consideration of the flexibility and supportability. In order to obtain the required gain, three stage capacitive feedback amplifiers were used to form the pre-amplifier. After the amplifier, there were two threshold programmable comparators which were used to detect the echo signal. The whole receiver was realized by using fully differential structure in order to reduce the noise effect.

As shown in Figure 5d, when it passes 229 system clock cycles from the transmission to the reception of the ultrasonic signal, the distance between the PMUT and the target is 22.9 mm.

## 4. System Evaluation

### 4.1. Power Consumption of the Chip

When the power supply voltage is 1.8 V, the chip can run in different modes by controlling the value of the flag, including CHIPON, TXOSCON, CP5_ON, CP32_ON and RXON. The chip has different power consumption in different working modes. By measuring the current consumption in different modes, the chip consumption was obtained, as shown in Figure 6b. Because there was a current leakage path, the idle and standby power consumption was high, which will be improved in the future.

### 4.2. Ranging Experiments and Results

Two PMUTs were connected to a fixed printed circuit board (PCB) through wire bonding as the transmitter and receiver, respectively. The experimental setup is shown in the Figure 6a. A reflector, which is a piece of smooth aluminum plate, was fixed on the measuring claw of a Vernier caliper with a resolution of 0.02 mm. A 1.8 V power supply was applied to power the transceiver chip. The oscilloscope was connected to the TX and RX terminals of the chip to monitor the transmitted electrical pulse and the received ultrasonic echo signal.

It was found that when the number of the TX pulses was 12, the PMUT can be fully oscillated. If the number continues to increase, the amplitude can no longer be increased significantly. Therefore, 12 TX pulses were used to stimulate the transmitter. The measurement results are shown in the Figure 6c. The blue line indicates the twelve electrical pulse excitation signals, and the brown line indicates the waveform of system clock with a frequency of 1.73 MHz. The purple line indicates the waveform of the receiver. Due to the crosstalk between the transmitter and the receiver, the receiver has an electrical signal fluctuation before the ultrasonic echo arrives, which can be filtered out by hardware isolation or filtering algorithms.

The experimental results of average of five measurements are summarized in Table 1. $n_{\mathrm{clk}}$ refers to the number of cycles of received voltage, L is the sensing distance. RX Vpp is the voltage amplified by the pre-amplifier, which is 500 times the actual received voltage. This data will determine the maximum test distance, because the longer the distance, the smaller the voltage of the reflected wave. Since the receiver can only sense certain limit of voltage, the sensing range is determined by the received voltage. 

The performance comparisons with previous works are listed in Table 2. For the same TOF method in [14], the range and accuracy of this work both have been improved due to the usage of more advanced fabrication process and customized circuits. Compared with the phase-shift method [23], the TOF method has a larger sensing range, but the accuracy is slightly lower, especially at short distances (e.g., <100 mm), while the phase shift method has extremely high accuracy. 

## 5. Conclusions

In this paper, we present a high-accuracy CMOS driven ultrasonic ranging system based on air coupled piezoelectric micromachined ultrasonic transducers (PMUTs) using time of flight (TOF). The mode shape and the time-frequency characteristics of PMUTs were firstly simulated and analyzed. PMUTs with frequencies of 97 kHz and 96 kHz were chosen as the transmitter and receiver, achieving both high range accuracy and ranging field. Based on the 0.18 μm CMOS process, the time to digital converter circuit simplifies the ranging method by system clock counting. The experimental results show that the accuracy of the ranging between 10 cm and 50 cm is 0.63 mm (one standard deviation), which is better than that of the previously reported rangefinder. The TOF based ranging system using PMUTs and CMOS circuit shows great application potentials in short-range communication, such as ultrasonic positioning, gesture recognition, medical assistance, etc. Future work will focus on the integration of CMOS and PMUTs, as well as applications based on ultrasonic rangefinders.

## Figures and Tables

**Figure 1 micromachines-12-00284-f001:**
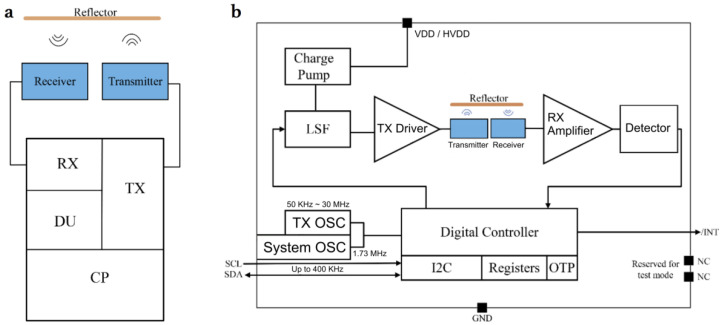
The block diagram of ultrasonic time of flight (TOF) ranging system using aluminum nitride (AlN) piezoelectric micromachined ultrasonic transducers (PMUTs). (**a**) One PMUT is a transmitter and the other is a receiver. (**b**) The function block diagram of the system.

**Figure 2 micromachines-12-00284-f002:**
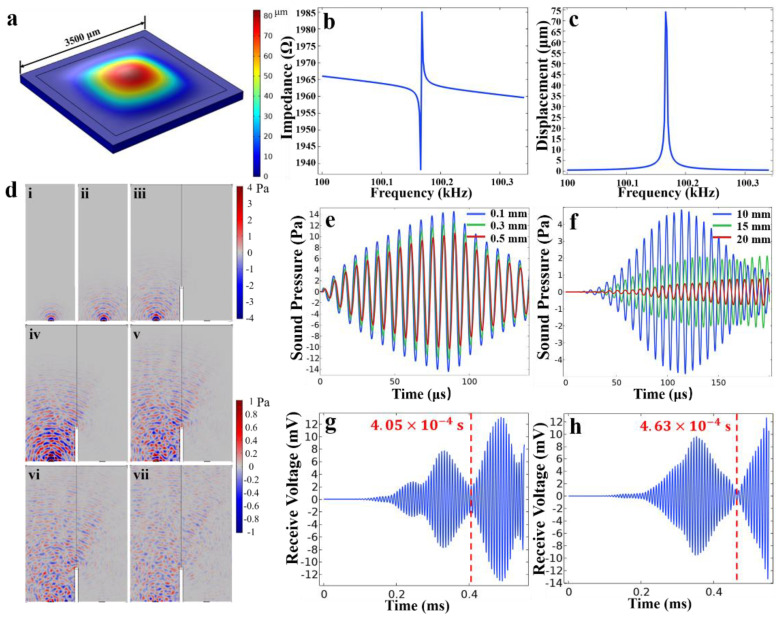
The finite element simulation results of PMUTs. (**a**) The first vibration mode diagram of PMUT. (**b**) Impedance and (**c**) displacement versus the frequency characteristics of the PMUT. (**d**) Propagation state of ultrasound when the time is (**i**) 30 μs, (**ii**) 60 μs, (**iii**) 90 μs, (**iv**) 120 μs, (**v**) 150 μs, (**vi**) 180 μs and (**vii**) 210 μs. The grayness in the picture is the air domain. The upper part of the air is the reflector. PMUTs are located on both sides of the bottom. The left is the transmitter and the right is the receiver. In the middle of the air is a baffle to prevent interference between the transmitter and receiver. (**e**) Near−field and (**f**) far−field sound pressure versus time, where the points of near−field and far−field sound pressure are 0.1, 0.3, 0.5 mm and 10, 15, 20 mm from the center of the film, respectively. The receiving voltage of the receiver when the sensing distance is (**g**) 7 cm and (**h**) 8 cm. The red dashed line is the arrival time of the reflected wave, and the specific time is marked on the left.

**Figure 3 micromachines-12-00284-f003:**
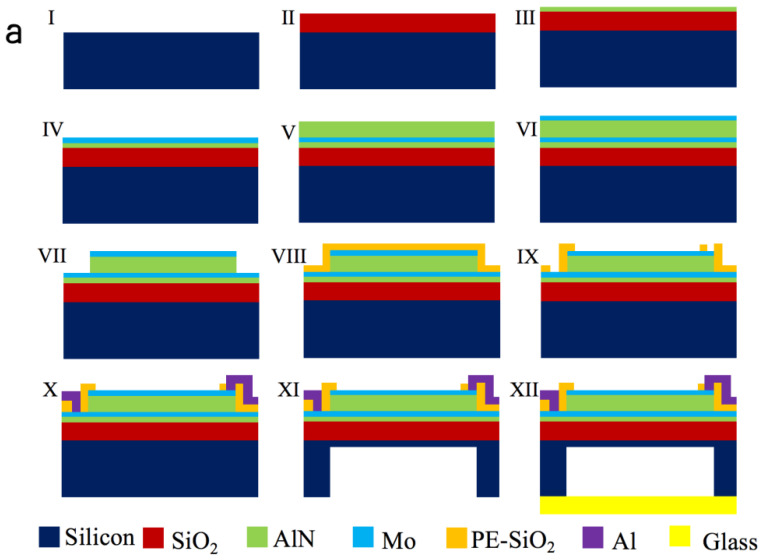
The fabrication and characterization of PMUTs. (**a**) The process flow of the air-coupled PMUT based on AlN. (**b**) Optical microscopy (OM) and SEM imaging of the PMUT (the adhesive layer is added for SEM imaging). (**c**) Analysis of impedance characteristics of transmitter (97 kHz), and (**d**) receiver (96 kHz).

**Figure 4 micromachines-12-00284-f004:**
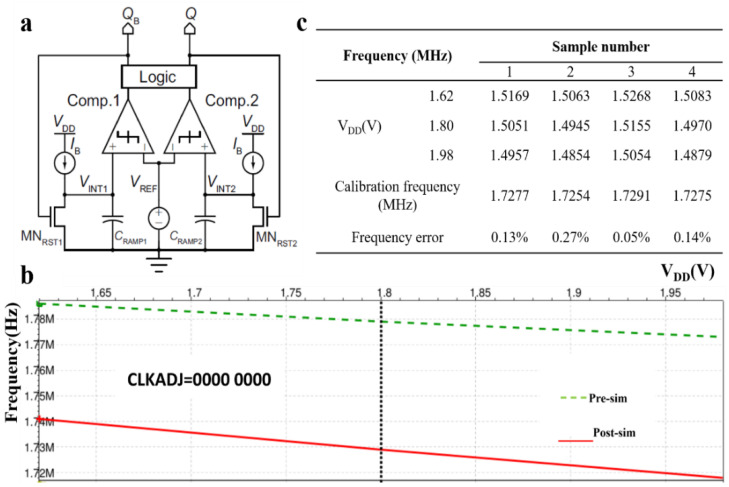
Realizing 1.73 MHz frequency of system (OSC) for TOF calculation. (**a**) The block diagram of the relaxation oscillator. (**b**) System frequency vs. power supply voltage by simulation, and (**c**) measurement results of the frequency vs. power supply.

**Figure 5 micromachines-12-00284-f005:**
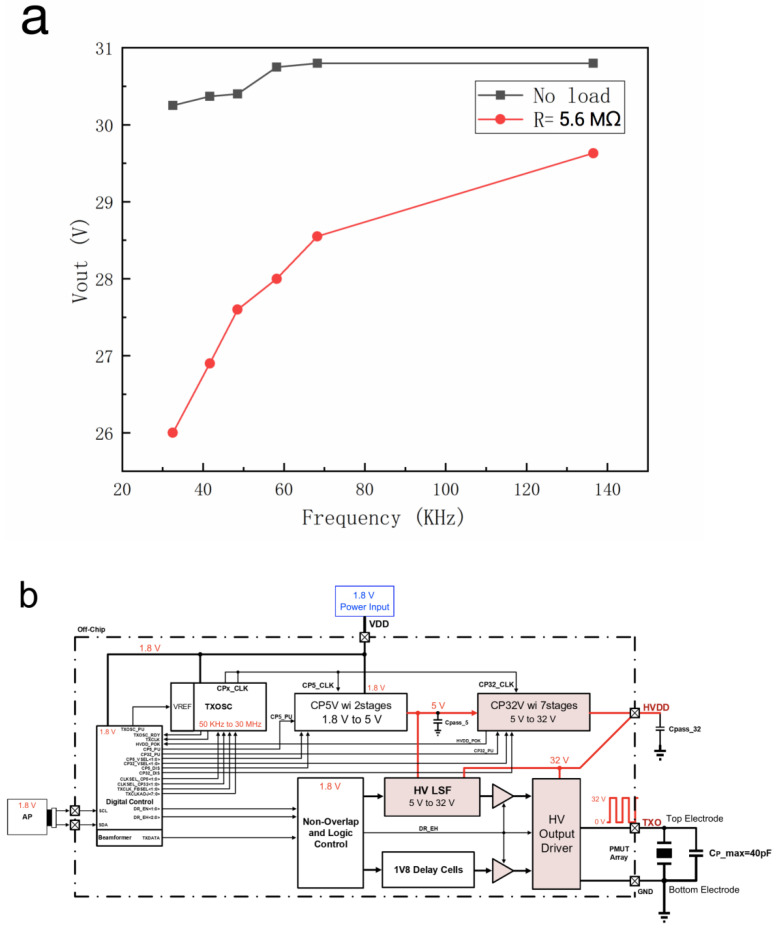
Charge pump, transmitter and receiver circuits. (**a**) The relationship between the output voltage of the charge pump circuit and the driving frequency. (**b**) The function block diagram of TX. Signal simulation of (**c**) the drive circuit and (**d**) the receiver circuit.

**Figure 6 micromachines-12-00284-f006:**
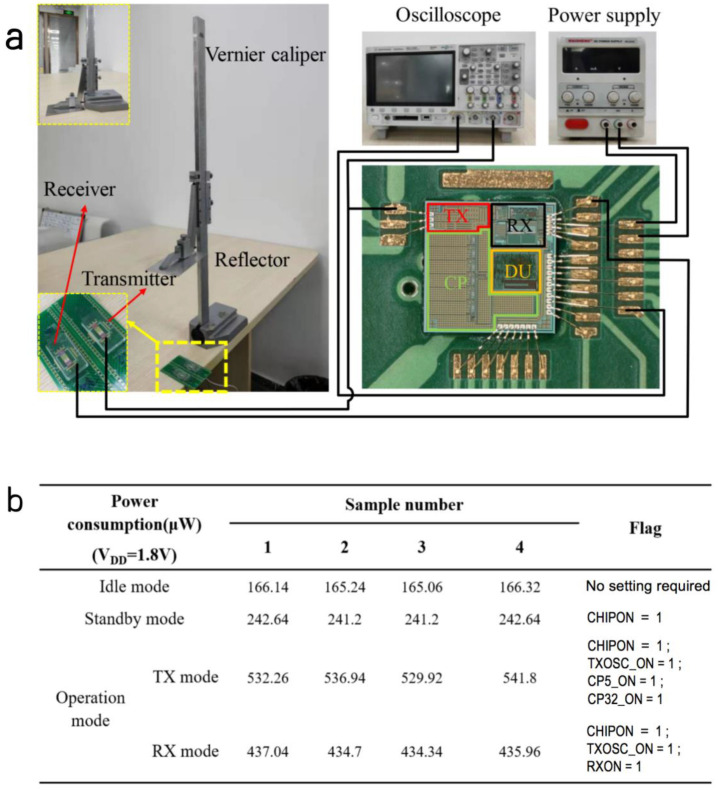
Measuring setup and results of ranging system. (**a**) The setup for range measurement (100–500 mm). (**b**) Power consumption of the chip in different working modes. (**c**) The experimental results of measuring waveforms. The distance from (**i**) to (**ix**) is from 10 cm to 50 cm in the increment of 5 cm.

**Table 1 micromachines-12-00284-t001:** The results of the ranging (100–500 mm).

Results	The Readings of Vernier Calipers (mm)	n_clk_	L (mm)	Range Errors (mm)	RX Vpp (mV)
1	100.24	1004	100.42	0.18	4012.8
2	150.12	1499	149.86	−0.26	2403
3	200.14	2008	200.8	0.66	2010.4
4	250.02	2505	250.48	0.46	1566.4
5	300.22	2997	299.68	−0.54	1262.2
6	350.78	3496	349.56	−1.22	952.4
7	400.08	3998	399.76	−0.32	752.4
8	450.22	4495	449.52	−0.7	584
9	500.32	4994	499.4	−0.92	343.2
Standard deviation σ (mm)				0.63	

**Table 2 micromachines-12-00284-t002:** Performance comparison with previous works.

Reference	Method	Transducers	Max Range	Range Error
[20]	Phase-shift sound	Si Thermal ultrasound	0.11 m	4 mm
[14]	TOF sound	AlN PMUT	0.45 m	3.9 mm (3σ)
[23]	Phase-shift sound	AlN PMUT	0.1 m/0.3 m	71.1 μm/1.82 mm (3σ)
This work	TOF sound	AlN PMUT	0.5 m	0.63 mm (3σ)

## Data Availability

All data, models, and code generated or used during the study appear in the submitted article.

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
