# Peer review of "A Novel Ultrasonic TOF Ranging System Using AlN Based PMUTs"

_micromachines, 2021, doi:10.3390/mi12030284_

Round 1
Reviewer 1 Report
In this study, Jin et al. presents a high-accuracy CMOS driven ultrasonic ranging system based on air-coupled AlN based piezoelectric micromachined ultrasonic transducers (PMUTs) using time of flight (TOF). The developed ranging system features excellent range, accuracy, and precision, highlighted by the capability of measuring at the range of 50 cm with an average error of 0.3 mm. The study was well-designed and the manuscript is well-written. It is therefore recommended for publication after minor revision to the english language (avoid using words like we).
Reviewer 2 Report
Please see attached file.

Reviewer 3 Report
Introduction is poorly focused on the novelty claimed in the introduction and reported in the manuscript. While it seems from the reviewed literature that AlN PMUTs are not the novel part as it is discussed they were widely investigated in the past years, it is necessary to evidence what is the novelty introduced by authors. If it is in the TOF algorithm/architecture, it should be better described.
L13-14 the sentence is not clear
L23-l30 please include references.Comparable performance is intended because AIN can withstand higher electric field?
Figure 1 is difficult to understand…It seems that the two transducers are included in the hardware and transmit/receive internally. Please clarify and/or improve the figure
Some acronyms were not defined (e.g. , SOI, TMSC etc. please check)
L98-99 “he CP will rise the 98 voltage of 1.8V to 5V and 32V to drive the DU” I supposed the driving and receiving circuit were analog. What us the reason of driving the DU with a charge pump. I found in general the description of the architecture largely fragmented. It results in lacks of general understanding of the novelty of the architecture.
L114-L119 Evaluation of resolution should be better described and justified.
All the Figures are poorly described din the caption. All the subcomponents should be better described.
Equation 3 should be checked. I also suggest to verify the reference as in some cases I was not able to find the information reported.
In L114 authors reported a sound velocity of346 m/s at 25 °C while in equation 4 using 25°C it results in a sound velocity much higher. Please check.
What are the results provided by Figure 4…please justify
Experimental validation should be better described with emphasis on the metrics used to evaluate the performances of the system. Moreover, a comparison with the most recent TOF algorithms will be of help (e.g. 10.3390/s18010089 10.3390/s20185042). Especially it should be interesting to compare the fundamental of the algorithm used (threshold, cross-correlation, Hilbert transform Etc.)
References could be expanded and updated.
Round 2
Reviewer 3 Report
N/A